# Prolonged Impact of Bisphosphonates and Glucocorticoids on Bone Mechanical Properties

**DOI:** 10.3390/ph18020164

**Published:** 2025-01-26

**Authors:** Alaa Mansour, Zaher Jabbour, Ammar Alsheghri, Amir Elhadad, Karla R. Berridi, Hanan Moussa, Jose Luis Ramirez-Garcialuna, Iskandar Tamimi, Sailer Santos dos Santos, Janet Henderson, Jun Song, Faleh Tamimi

**Affiliations:** 1Private Dental Practice, Ottawa, ON K1P 5Z9, Canada; alaamans@buffalo.edu; 2School of Dentistry, University of California, Los Angeles, CA 90095, USA; 3Mechanical Engineering Department, King Fahd University of Petroleum and Minerals (KFUPM), Dhahran 31261, Saudi Arabia; ammar.sheghri@kfupm.edu.sa; 4Interdisciplinary Research Center for Biosystems and Machines, King Fahd University of Petroleum and Minerals (KFUPM), Dhahran 31261, Saudi Arabia; 5College of Dental Medicine, QU Health, Qatar University, Doha P.O. Box 2713, Qatar; aelhadad@qu.edu.qa; 6Bone Engineering Labs, Injury Recovery Repair Program, Research Institute McGill University Health Centre, Montreal, QC H4A 3J1, Canada; karla.rangel-berridi@mail.mcgill.ca (K.R.B.); jose.ramirezgarcialuna@mail.mcgill.ca (J.L.R.-G.); janet.henderson@mcgill.ca (J.H.); 7Experimental Surgery, Faculty of Medicine, McGill University, Montreal, QC H3A 0G4, Canada; 8Faculty of Dentistry, McGill University, Montreal, QC H3A 0G4, Canada; hanan.moussa@mail.mcgill.ca; 9Faculty of Dentistry, Benghazi University, Benghazi 435C W26, Libya; 10Orthopedic Surgery Department, Regional University Hospital of Málaga, 29010 Málaga, Spain; isktamimi80@yahoo.com; 11Department of Chemistry, Federal University of Santa Maria, Santa Maria 97105-900, Brazil; sailer.santos@ufsm.br; 12Department of Mining and Materials Engineering, McGill University, Montreal, QC H3A 0G4, Canada; jun.song2@mcgill.ca

**Keywords:** bone collagen, bisphosphonates, glucocorticoids, biomechanics, drug holiday

## Abstract

**Background:** This study aimed at investigating the prolonged effects of glucocorticoids and bisphosphonates on bone. **Methods:** Six-to-eight-month-old skeletally mature male Sprague Dawley rats were randomized to receive a cancer therapy combination of zoledronic acid (ZA = 0.13 mg/kg) and dexamethasone (DX = 3.8 mg/kg) (treatment group, *n* = 10) or sterile phosphate buffer saline solution (control group, *n* = 10). The rats received weekly intraperitoneal injections for 8 weeks, which were stopped 6 weeks before euthanasia. Mineralized bone samples were characterized by three-point bending tests, micro-CT imaging, X-ray diffraction (XRD), thermogravimetric analysis (TGA), and differential scanning calorimetry (DSC). Bone collagen was assessed using tensile tests on the demineralized bones and attenuated total reflectance Fourier transform infrared (ATR-FTIR) spectroscopy on mineralized and demineralized bones. **Results:** The samples in the treatment group showed increased tibial cortical thickness, mineral crystal size, and toughness. Analyses of demineralized tibiae revealed decreased collagen tensile strength in the experimental group. The spectroscopic and TGA/DSC analyses showed that the ZA + DX treatment increased the collagen amide I 1660/1690 cm^−1^ area ratio and collagen denaturalization temperature, indicating a higher level of collagen cross-linking. **Conclusions:** Bisphosphonates and glucocorticoids led to prolonged changes in the mechanical properties of bone as a result of increased cortical thickness, increased crystal size, and the deterioration of collagen quality.

## 1. Introduction

The combined administration of bisphosphonates and steroids is often used in the treatment of cancer (e.g., prostate cancer, breast cancer, multiple myeloma, lung cancer, etc.) and autoimmune diseases (e.g., rheumatoid arthritis, lupus erythematosus, psoriasis, Sjögren’s syndrome, temporal arteritis, ulcerative colitis, etc.) [1]. Bone metastases can be osteoblastic (e.g., prostate cancer in males), osteolytic (e.g., breast cancer in females and multiple myeloma in both males and females), or mixed. As a result, bisphosphonates have been widely used in cancer treatment to reduce the number of bone-related events and preserve the bone integrity. Glucocorticoids are also used extensively in the treatment of cancer as an adjuvant therapy to reduce the side effects of cancer medications.

The effect of glucocorticoids on bone is complex and includes more than one mechanism [2]. Glucocorticoids have an impact on mesenchymal cells, leading to their differentiation into adipocytes, which decreases osteogenesis. In addition, glucocorticoids have a direct impact on osteoblasts by affecting their maturation, lifespan, and function. As a result, bone formation is decreased and results in a net bone loss and osteoporosis [2].

Bisphosphonates are a class of drugs used to treat bone-related conditions, such as osteoporosis and metastatic cancer [3]. These drugs are effective in increasing bone mineral density and decrease the risk of osteoporotic fractures [4]. Bisphosphonates are often used to treat certain cancers because they can have a direct anti-tumor cell effect that inhibits tumor formation and growth [3], and they prevent pathological bone fracture caused by steroid-induced osteoporosis [5]. However, the administration of bisphosphonates was linked to atypical bone fractures in the femoral shaft due to repetitive stress, microdamage, and fatigue [6].

The risk of atypical bone fractures increases with age [4,7], which seems to indicate that it might be related to bone protein aging in terms of loss of elasticity and increased glycosylation [7]. Collagen cross-linking plays a key role in defining the biomechanical properties of bone, including bone mineralization and the formation of microdamage [7]. These cross-links can be enzymatic immature divalent cross-links, mature trivalent cross-links, and glycation- or oxidation-induced non-enzymatic cross-links (advanced glycation end products). While enzymatic cross-link formation is beneficial for bone biomechanics, AGE cross-linking can have a detrimental effect on the material properties, and it could cause senescence from an older matrix [8]. 

To decrease the risk of atypical bone fracture, recent studies highlighted the positive impact of bisphosphonates discontinuation, even after a short period of time [4,6,9,10,11]. Strom et al. suggested that the risk of atypical fracture decreased by 60% within the first 6 months of bisphosphonates’ discontinuation in patients who took oral bisphosphonates for osteoporosis for more than 12 months [11]. Black et al. also reported rapid decline in the risk of atypical fracture after a bisphosphonate holiday [4]. However, these studies evaluated the impact of bisphosphonates’ discontinuation in patients treated for osteoporosis with no data available on the impact of bisphosphonate holidays in cancer patients. As a result, much controversy still exists about the long-term impact of high doses of bisphosphonates on bone in cancer patients, especially when combined with other cancer medications such as corticosteroids. Therefore, the aim of this study was to investigate the long-term effects of cancer doses of bisphosphonates and corticosteroids on bone.

## 2. Results

All the animals completed the study (*n* = 10 in each group). The combined treatment with ZA + DX led to a statistically significant increase in the average bone cortical thickness, absolute maximum force, and toughness (energy to fracture per unit volume) when compared to the control (Table 1). Higher toughness values were positively correlated with an increased cortical thickness measured via micro-CT (Pearson correlation r = 0.56, *p* = 0.045). No significant difference in the stiffness, elastic modulus, flexural strength, or flexural strain was observed in treated bones when compared with the controls (Table 1). Five control and two treated samples were excluded from these analyses because they experienced a very oblique fracture resembling a greenstick bone fracture. This guaranteed an accurate comparison for the cortical thickness of fractured bone samples for a cross-section taken at 0.5 mm from the fracture line, which was located in the middle of the bone for the case of a normal fracture.

The stress–strain curve for the mineralized samples in the treatment group exhibited a higher level of resistance to fracture after reaching the maximum stress compared to the control curve (Figure 1a). This indicated increased toughness in the treated bones. The odds ratio for having an oblique (greenstick) fracture in the control group was four times higher than in the treated group (OR = 4.0; 95% CI 0.5–29). However, this difference did not reach a statistically significant level (*p* = 0.17, Fisher’s exact test).

Unlike the mineralized samples, the assessment of the demineralized samples revealed that the tensile strength for the control group was significantly greater than the treated group (Table 1). The typical force–displacement diagrams for the control and treated demineralized bones showed almost linear elastic curves (Figure 1b). However, as the load increased, the specimens reached a point of yielding and plastic deformation. After that, fractures occurred at a strain of about four percent for the control group and three percent for the treated group. However, when comparing the maximum tensile force and stiffness between the control and treated demineralized bone samples, the difference did not reach a statistically significant level (*p* = 0.167). Nevertheless, statistically significant nonparametric correlations were observed between the maximum tensile force of the demineralized group and the three-point bending maximum force of the mineralized group in both groups (control group’s Spearman correlation coefficient = 1.00, *p* = 0.01, R^2^ = 0.76; treated group’s Spearman correlation coefficient = 1:00, *p* = 0.01, R^2^ = 0.87 (Appendix A)). Two treated demineralized samples and one control sample were excluded from the analysis because they fractured near the grip.

The PXRD patterns of the treated and control bone samples showed that the combined bisphosphonate and glucocorticoid treatment resulted in greater diffraction peaks for the main exposed planes of hydroxyapatite crystals (Figure 2). In addition, although a larger crystal size was observed in the treated group (195.1 ± 24, 53.6 ± 3.4, and 102.3 ± 17.5 Å) compared to the control group (186.5 ± 30.6, 50.9 ± 1.5, and 85.6 ± 16.5 Å) at the (002), (211), and (310) planes, respectively, the difference was statistically significant only for the 211 plane (Table 2).

The representative ATR-FTIR spectra of the control and treated bone from the mineralized and demineralized groups are shown in Figure 3. The peak assignments corresponding to the protein and mineral constituents were based on Movasaghi et al. [12]. The ATR-FTIR spectra in the region from 1000 to 1200 cm^−1^ is dominated by mineral bands (ν_1_ and ν_3_ stretching modes of PO_4_^3−^) and contain collagen bands at 1238 cm^−1^ (amide III band: N-H in-plane bending coupled with C-N stretching, also C-H and N-H deformation vibrations), the amide II band near 1544 cm^−1^ (a combination of the C-N stretch and N-H bending modes), and the amide I at 1642 cm^−1^ (peptide C=O stretching weakly coupled with C-N stretching and N-H bending), which represents the organic matrix of bone since the vast percentage of the protein content is collagen type I [13,14]. This region corresponds to a wavelength of 1600–1720 cm^−1^, which is highlighted between the two yellow vertical lines in Figure 3a,c. It is highly sensitive to the conformational changes in the secondary structure [12]. The bands at 1380 and 1450 cm^−1^ correspond to absorptions from CH_2_ and CH_3_ wagging. The broad band at 3321 cm^−1^ corresponds to amide A (N-H stretching vibration) and to the O-H component, which confirms the active participation of water in the collagen molecule [15,16].

The ZA + DX treatment resulted in significant changes in both the mineralized and demineralized bone samples. For the mineralized samples, significant changes were observed in the area of the peaks at wavenumbers 1046, 1086, 1647, 2129, 2894, and 3323 cm^−1^, the full width at half maximum (FWHM) at 3323 cm^−1^, the center at 1740, 1923, and 2974 cm^−1^, and the height at 3323 cm^−1^. The demineralized samples also presented significant changes in the area at 1080, 1239, 1546, 1647, 2894, and 2973 cm^−1^, the FWHM at 1080, 1647, 2925, and 2973 cm^−1^, the center at 1080, 1160, 1239, 1334, 1380, and 2973 cm^−1^, and the height at 1239, 1334, 1380, 1451, 1546, and 1647 cm^−1^ (Appendix A). The changes in peaks represent changes in collagen and phosphates, especially in the demineralized samples.

The amide I spectral region was resolved into its underlying components. Of these components, the relative percent area ratio of two sub-bands at ∼1660 cm^−1^ and ∼1690 cm^−1^ was related to collagen cross-links. By the curve-fitting of the amide I band, we examined the collagen chemistry (Appendix A). It has been shown that the 1660/1690 cm^−1^ ratio (a marker of collagen cross-links) correlates with the relative amounts of nonreducible (mature) to reducible (immature) types of cross-links [17,18]. The ratio was calculated using a peak-fitting analysis, with the three main peaks in the amide I band [16], and it varied significantly (*p* < 0.05) among groups. Hence, the spectroscopic analysis of the samples revealed an increase in the 1660/1690 cm^−1^ area ratio of the treated group compared with the control group for both the mineralized and demineralized samples, indicating a higher level of cross-linking in the collagen of the treated samples.

Differential scanning calorimetry and thermogravimetric analyses (DSC-TGA) were used to determine the transformation temperatures and the mass loss of the control and treated samples (Figure 4). In both groups, the main loss was presented in three different temperature ranges given by 30–189 °C, 189–293 °C, and 293–700 °C. The first substantial weight loss occurred in the 30–189 °C temperature range, corresponding to 10.80 ± 0.54 wt% in the control group compared with 4.33 ± 0.21 wt% in the treated group. This loss could be attributed to the evaporation of the physiosorbed and solvate water molecules in the material as detected by the two endothermic peaks on the DSC graph at 141 °C and 171 °C in the control group and at 144 and 199 °C in the treated group [19]. This indicates that treatment with bisphosphonates and corticosteroids could reduce the water content.

The second weight loss occurred in the range of temperatures from 189 to 293 °C, corresponding to 11.70 wt% loss in the control group compared with 15.30 wt% loss in the treated group (Figure 4). This event could correspond to the distortion of the triple-helix structure of collagen in the bone matrix, which is dependent on the degree of hydration of the material and the elimination of the attached free water of the hydroxyapatite matrix. The last substantial weight loss occurred at 293–700 °C, presenting a maximum change in mass loss corresponding to 50.2 wt% loss in the control samples and 28.9% in the treated samples. This loss could be attributed to the decomposition of organic components in bone accompanied by the volatilization of gaseous compounds from this process [20].

## 3. Discussion

This study showed that the ZA + DX treatment increased toughness (i.e., the energy to fracture per unit volume) of mineralized bone due to the increased cortical thickness. However, the treatment also increased collagen cross-linking, as revealed by the spectroscopic analysis of the amide I band and decreased the collagen tensile strength, indicating a loss of ductility. Furthermore, the combined administration of bisphosphonates and glucocorticoids reduced the water content and increased the crystal size of bone minerals. These findings could be consistent with the observation that control samples had more oblique (greenstick) fractures than experimental samples, although the difference did not reach a statistically significant level. More samples in our experimental group appeared to have transverse or short oblique fracture configurations, which is one of the major features of atypical fractures [21]. Although the nature of bone fractures is a secondary outcome in our study, it is consistent with the literature indicating that bisphosphonates and glucocorticoids are risk factors for atypical bone fractures [4]. Nevertheless, in the same study, it was reported that the risk of atypical fracture decreases rapidly after bisphosphonates’ discontinuation, which could explain why the type of fracture in our study did not reach a statistically significant level. The incidence of atypical fractures could be also affected by gender [22]. Males had more atypical fractures than females in individuals younger than 49 years old. However, females aged 80–90 years represented most cases of atypical fractures in individuals older than 50 [22].

The three-point bending test results showed an increase in the toughness and maximum force of treated tibiae compared to the control (Table 1). These increments could be related to a toughening mechanism, which was significantly correlated with the increased cortical thickness of the tibia due to treatment. As a result, the tissue density was increased, requiring more energy per unit volume to cause bone fractures. Since the three-point bending test revealed a similar elastic modulus and flexural strength in both groups (Table 1), it appears that the increased cortical thickness in the treated bones compensated for the loss of collagen’s tensile strength and ductility (Appendix A). Therefore, it seems that collagen is important for bone strength in healthy control rats, whereas the crystal size is important for bone strength in the experimental rats. Our three-point bending test results of the control group are comparable to previous reports [23]. This was also true for the maximum tensile force values obtained from the tensile testing of demineralized bone samples, which were comparable to the values previously reported in the literature [24].

One of the main reasons for the reduction in collagen ductility might be the increase in cross-linking. The ATR-FTIR results showed changes in peaks related to collagen, especially in the demineralized samples. More specifically, changes in the amide I band could be associated with increased cross-linking due to ZA + DX administration, as demonstrated by the spectroscopic analysis. The drug combination seems to interrupt the remodeling activity and result in the deterioration of the collagen integrity, which affects its mechanical performance. In addition, the TGA curve of the treated sample (Figure 4) showed that endothermic peaks shifted to higher temperatures (144, 199, 314, 454, and 508 °C) when compared with those in the control group (141, 171, 309, 411, and 506 °C). These findings indicate an increase in the thermal stability of the treated bone samples due to the higher levels of cross-linking of the collagen network. Generally, the improvement in thermal stability is an indication of cross-linking [25]. Similarly, the residual mass at 700 °C exhibited the same trend with the treated bone samples having higher decomposition temperatures and a greater residual mass than the control samples. In addition, the significantly greater weight loss that occurred in the control group in the 30–189 °C temperature range (Figure 4) could be attributed to the evaporation of the physiosorbed and solvate water molecules. This would indicate that the bisphosphonate treatment might result in a reduction in bone hydration. The DSC/TGA results are along the same lines of the experimental tensile test and the spectroscopic analysis, proving that treatment with this drug combination increased collagen cross-linking and decreased the water content, causing a reduction in bone ductility.

Due to the increased crystal size in the treated bone samples compared with control bone samples, it is expected that the combined administration of ZA + DX resulted in a reduction in the bone fracture toughness. As the crystal size increases, a fraction of the crystal boundary phase is decreased and thereby a lower amount of energy is absorbed during crack propagation through the crystal boundaries [26,27]. The increase in the crystal size, as shown from XRD results (Figure 2), and the reduction in collagen quality with treatment may offer a possible explanation for the incidence of bone fracture in individuals receiving bisphosphonates/glucocorticoids combination therapy. A potential recommendation to avoid a deterioration of bone integrity in patients could be to use alternative osteoporotic treatments, such as teriparatide, that stimulate osteoblasts and increase bone formation instead of inhibiting osteoclasts and decreasing bone resorption. However, further research should be performed to analyze the impact of such treatments on patients with long-term use of glucocorticoids [28].

Although the fracture risk appears to decrease significantly after oral bisphosphonates’ discontinuation [4,6,9,10,11], the antiresorptive effect of bisphosphonates seems to remain in the bone for an extended period of time even after a drug holiday [10]. Bisphosphonates are incorporated in bone and reduce the lifespan and activity of osteoclasts, which decreases bone resorption [29]. Since bisphosphonates are incorporated within the bone structure, when bone containing bisphosphonates is resorbed, bisphosphonates are released locally and systematically and become available to bind again on other bone surfaces [29].

This study is limited to two groups, one experimental group which received the drug combination followed by a drug holiday and one control group which received PBS. Additional studies with more experimental groups are needed to determine the impact of bisphosphonates and corticosteroids separately on bone. Future studies with a wider range of treatments, doses, and endpoints are needed to further improve our knowledge on the long-term effects of the combined therapy of bisphosphonates with glucocorticoids. Even though our study was designed with a sufficient sample size to address our main objectives, some secondary observations, such as those of oblique greenstick fractures, led to samples being excluded from the study. This approach ensures an accurate and conservative interpretation of the results. Larger sample sizes are needed to fully explore the observation of greenstick fractures. Furthermore, our study did not include cellular or histomorphometric analyses of bone. These analyses were previously described in the literature [30,31,32] and are important considerations for the interpretation of the results.

## 4. Materials and Methods

### 4.1. Animal Model

Animal procedures were conducted in accordance with a protocol approved by the Facility Animal Care Committee of McGill University (AUP-7815) and in keeping with the guidelines of the Canada Council on Animal Care. Six-to-eight-month-old skeletally mature male Sprague Dawley rats (Charles River Laboratories, Senneville, QC, Canada) were used for the current experiment. Each rat was maintained in a single cage, in a controlled environment at 22 °C with 12 h light/dark cycles. Animals were fed on ad libitum during the entirety of the experiment.

Animals were randomized to either control group (*n* = 10) or treatment group (*n* = 10). Animals in the control group received weekly injections of sterile phosphate buffer solution (PBS), whereas animals in the treatment group received weekly intraperitoneal injections of zoledronic acid (ZA = 0.13 mg kg^−1^) (Sigma Aldrich, Oakville, ON, Canada 1724827-150MG) plus dexamethasone (DX = 3.8 mg kg^−1^) (Sigma Aldrich D2915- 100MG) (i.e., ZA + DX). These combined high doses of glucocorticoids and bisphosphonates are equivalent to human doses prescribed to cancer patients and have shown to induce pathological changes in the jawbone of rats, resulting in osteonecrosis-like lesions [33,34]. After 8 weeks of treatment, all rats received drug holiday for 6 weeks. Euthanasia was then performed by CO_2_ asphyxiation under inhaled anesthesia. The tibiae samples were carefully dissected to be free of soft tissue and fixed for 24 h in 70% alcohol. The bones were then rinsed 3× with sterile PBS and stored at 4 °C for further analysis.

### 4.2. Mechanical Bending Test

From each animal, one tibia was used for mechanical bending test of mineralized structure and the other for tensile strength testing of demineralized organic matrix. Biomechanical testing of animal long bones such as tibia offers a reliable and feasible method to investigate the impact of pharmacological agents on bone properties compared with non-invasive radiography or serum bone marker assays [35].

A three-point bending test was carried out on the rat tibia to evaluate the mechanical properties using Mach-1 (Biomomentum Inc., Laval, QC, Canada). Ten mineralized bone samples were tested in each of the control and treatment groups. The tibia was positioned on the two supports on the medial aspect. The loading pin was pushed down against the tibia at a speed of 1 mm/min until fracture. The flexural strength  (σ), bending elastic modulus (E), stiffness (K), and flexural strain (ε) were calculated from the load–displacement curve recorded by the machine as follows [36]:(1)$$\sigma =\frac{F_{ultimate}Lc}{4I}$$(2)$$E=\frac{F_{elastic}}{d_{elastic}}\frac{L^{3}}{48I}$$(3)$$K=\frac{F_{elastic}}{d_{elastic}}$$(4)$$\varepsilon =\frac{12cd_{ultimate}}{L^{2}}$$
where $F_{elastic}$ and $d_{elastic}$ are the force and displacement of the loading pin, respectively, measured by the machine in the elastic region of the stress–strain curve, $F_{ultimate}$ and $d_{ultimate}$ are the force and displacement recoded by the machine when the maximum force is reached, L is the distance between the two supports, which was fixed at 20 mm, I is the second moment of area of the bone cross-section in the vicinity of the fracture line, and c is the maximum chord length from the centroid axis of the bone cross-section as shown in Figure 5. Toughness was measured by calculating the area under the stress–strain curve until the point of fracture. The second moment of area and the centroid of the bone cross-section were calculated using the methods of Fiji (https://imagej.net/downloads, accessed on 17 January 2025) [37] and Bone J (https://bonej.org, accessed on 17 January 2025, National Institute of Health, Bethesda, MD, USA) [38].

### 4.3. Tensile Strength Test

Tensile pull-out tests were conducted on completely demineralized bones from both groups to test the mechanical properties of collagen. Tibial bone samples obtained from euthanized rats were demineralized for tensile testing of the collagen matrix. Both the control and treated groups consisted of 10 demineralized test specimens each. Mechanical tester Mach-1 (Biomomentum Inc., Laval, QC, Canada) was used with a speed of 2 mm min^−1^ [24], corresponding to a strain rate of approximately 0.2% s^−1^ as previously described [39]. The tensile force was measured using a 250 N load cell [24]. All tests were conducted at room temperature. A standard recording of the load–displacement curve was made by the machine. Sandpapers were attached between the grips and bone ends to minimize the slippage of the bone during the test. All samples were marked with two horizontal lines to help measure the deformation of the collagen at failure as previously reported [24]. The length measurement was made using a Vernier caliper. The strain at failure was then calculated by dividing the deformed length by the initial length between the two marks. Due to difficulty in measuring the exact cross-sectional area of demineralized bone samples, their tensile strength was calculated by dividing the tensile force of each demineralized bone sample by the cross-sectional area of the corresponding mineralized bone sample.

### 4.4. Micro-Computed Tomography (Micro-CT) Examination

Fractured mineralized bone samples were examined by micro-CT (Sky-Scan1172; Bruker, Kontich, Belgium). Scanning parameters were selected as follows: camera resolution of 6.0 µm, 25 kV, a 0.5 rotation step, exposure time of 2655 milliseconds, and aluminum filter. All samples were scanned using the same parameters, followed by the reconstruction procedures and the subsequent CT analysis using CTAn software (https://www.bruker.com/en/products-and-solutions/preclinical-imaging/micro-ct/3d-suite-software.html, accessed on 17 January 2025, Skyscan, Kontich, Belgium). A coronal view of the bone samples was used for all samples. To visualize the bone as close as possible to the site of fracture, the region of interest (ROI) was determined at 0.5 mm from the fracture line and then an appropriate binary threshold was chosen. Images from all examined bone samples were taken 0.5 mm from the fracture site.

### 4.5. Powder X-Ray Diffraction (PXRD)

Crystallography of powder prepared from mineralized bone of the treatment and control groups was examined using a D8 Discovery X-ray Diffractometer (Bruker-AXS, Karlsruhe, Germany) equipped with a copper source and operated at 40 kV and 40 mA. Diffractograms were obtained by measuring three subsequent frames with 300 s per frame (λ = 1.54056 Å), and data analysis was performed using DIFFRAC plus EVA software (https://www.bruker.com/en/products-and-solutions/diffractometers-and-x-ray-microscopes/x-ray-diffractometers/diffrac-suite-software/diffrac-eva.html, accessed on 17 January 2025, Bruker AXS, Karlsruhe, Germany). The crystal size (Å) was estimated from X-ray diffraction peaks using Scherrer’s equation as follows:(5)$$crystal\;size=\frac{K\lambda }{B\;cos\theta }$$
where *K* is a constant with a value of 0.89, λ is the wavelength of the used X-ray, *B* is the full width at half maximum (rad) of the relevant peaks (002, 211, and 310), and θ is the Bragg angle in rad.

### 4.6. Thermogravimetric Analysis (TGA) and Differential Scanning Calorimetry (DSC)

Thermogravimetric analysis (TGA) and differential scanning calorimetry (DSC) were performed with TGA module of Thermal Analysis System (TGA/DSC 1 from Mettler Tolledo), Columbus, OH. USA An amount of 5 mg of dried mineralized bone samples was placed in an open pan (Al_2_O_3_) attached to a microbalance. The samples were heated under nitrogen flow from 30 to 700 °C at a rate of 10 °C min^−1^.

### 4.7. Attenuated Total Reflectance Fourier Transform Infrared Spectroscopy (ATR-FTIR)

To identify the functional groups within the control and treated bone samples, attenuated total reflectance Fourier transform infrared (ATR-FTIR) spectroscopy of mineralized and demineralized samples was performed using a Bruker Tensor 27 IR spectrometer (Bruker Optics Ltd., Coventry, UK). This spectrometer is equipped with a DTGS (deuterated triglycine sulfate pyroelectric) detector with an accumulation of 128 scans in the range of 400−4000 cm^−1^ at a resolution of 4 cm^−1^. The upper part of the tibia samples was analyzed by ATR-FTIR.

Curve-fitting of the FTIR spectra was carried out using Origin 2019b. The initial position and type (Gaussian) of underlying bands were determined through the second derivative and difference spectroscopy. Once the curve-fitting process converged, the output of the analysis was expressed as peak position and relative % area.

### 4.8. Statistical Analysis

Data are presented using mean and SD. Differences between groups were assessed using the Mann–Whitney U test with *p* < 0.05. Bone samples with oblique (greenstick) fractures were excluded from all properties listed in Table 1. This guaranteed an accurate comparison for the cortical thickness and ensured more conservative interpretation of results. Statistical analyses were performed using SPSS Statistics 17 (IBM SPSS Inc, Chicago, IL, USA) software.

## 5. Conclusions

Our data suggest that the combined administration of bisphosphonates and glucocorticoids led to prolonged changes in the mechanical properties of bone as a result of an increased cortical thickness, increased crystal size, and the deterioration of collagen quality. Therefore, within the limitations of our study, the outcomes could indicate that the administration of bisphosphonates to patients treated with high doses of corticosteroids could improve the mechanical properties of bone. This effect could persist even after a discontinuation of the medications.

## Figures and Tables

**Figure 1 pharmaceuticals-18-00164-f001:**
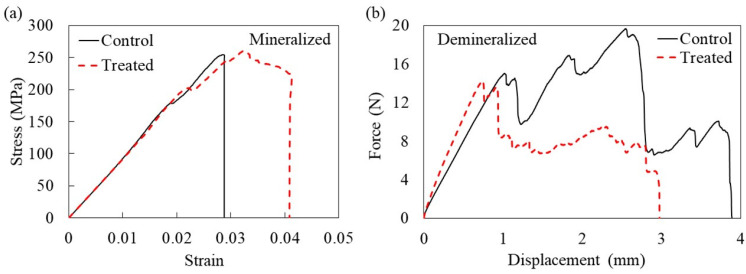
Samples for (**a**) stress–strain curves from 3-point bending test of control and treated mineralized bone; and (**b**) force–displacement curves from tensile testing of control and treated demineralized bone.

**Figure 2 pharmaceuticals-18-00164-f002:**
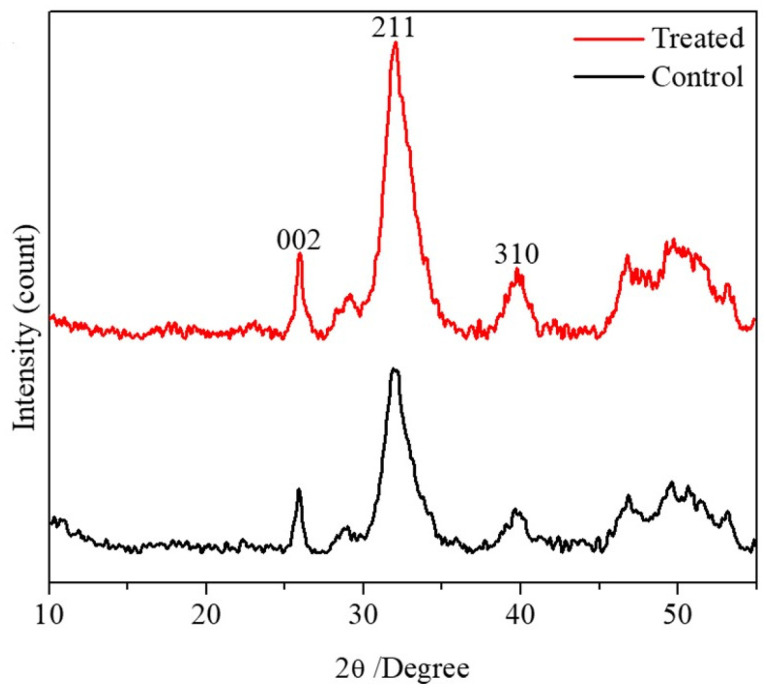
PXRD patterns of treated and control bone samples, showing changes in 211 planes in the treated group.

**Figure 3 pharmaceuticals-18-00164-f003:**
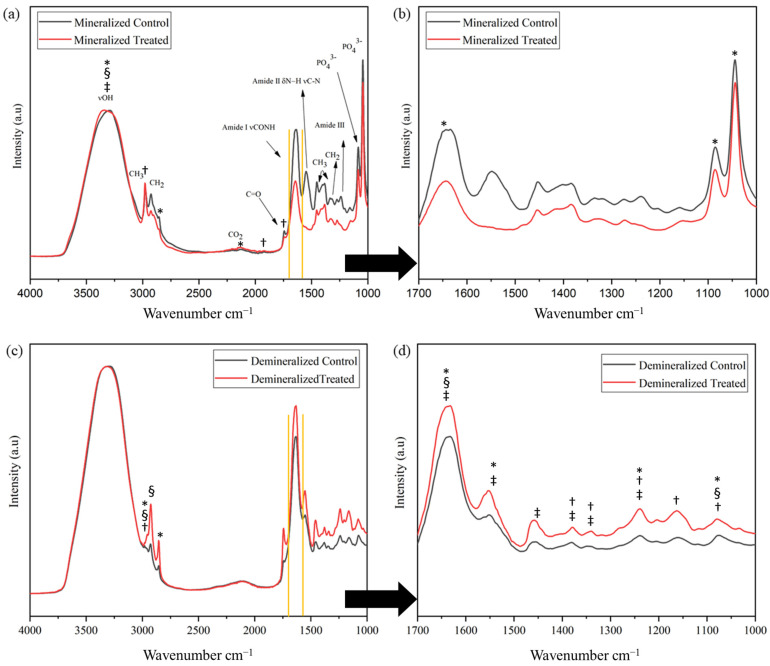
ATR-FTIR spectra of control (black) and treated (red) bone groups for mineralized bone samples. (**a**) Wavenumber between 1000 and 4000 cm^−1^; (**b**) Wavenumber between 1000 and 1700 cm^−1^. ATR-FTIR spectra of control (black) and treated (red) demineralized bone groups. (**c**) Wavenumber between 1000 and 4000 cm^−1^; (**d**) Wavenumber between 1000 and 1700 cm^−1^. Symbols indicate statistically significant differences between peaks * areas, § full width at half maximum (FWHM), † center, ‡ maximum height. Yellow vertical lines in (**a**,**c**) correspond to wavelength of 1600–1720 cm^−1^, which represents the region of the amide I band of tissue proteins.

**Figure 4 pharmaceuticals-18-00164-f004:**
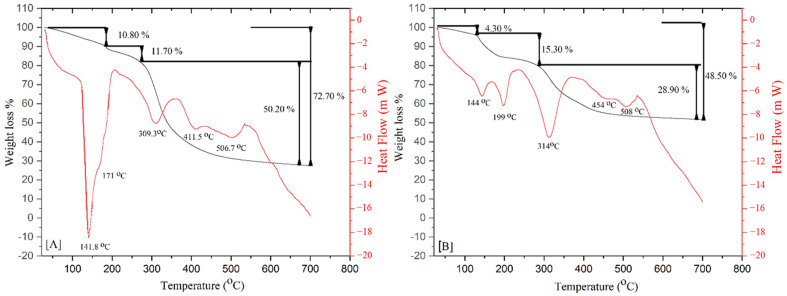
Typical DSC and TGA curves of [**A**] control and [**B**] treated bone samples.

**Figure 5 pharmaceuticals-18-00164-f005:**
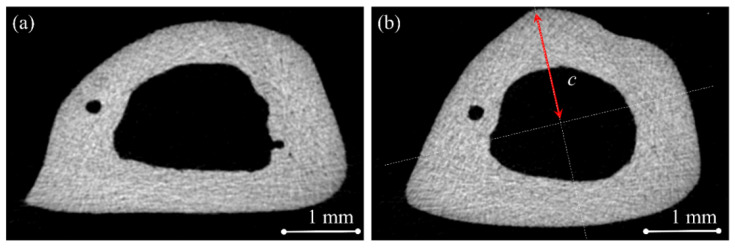
Cross-sectional photograph of the bending fracture site showing the centroid of the cross-section and the maximum chord length from the centroid (c) axis for (**a**) control and (**b**) treated groups; obtained using methods of Fiji and Bone J.

**Table 1 pharmaceuticals-18-00164-t001:** Summary of three-point bending test of mineralized samples and tensile pull-out tests of demineralized samples.

Property Mineralized Samples	Control (*n* = 5)	Treatment (*n* = 8)	*p*-Value
Bone length (mm)	49.73 ± 2.7	49.26 ± 1.76	0.284
Max chord length from centroidal axis (mm)	1.64 ± 0.1	1.63 ± 0.1	0.943
Cross-sectional area of cortical bone (mm^2^)	6.37 ± 0.4	6.94 ± 0.67	0.065
Average cortical thickness (mm)	0.78 ± 0.06 *	0.88 ± 0.04 *	0.006
2nd moment of area (mm^4^)	4.51 ± 0.75	4.85 ± 1.01	0.512
Max force (N)	137 ± 11.6 *	157 ± 11.5 *	0.030
Stiffness (N mm^−1^)	278 ± 107	290 ± 20	0.354
Flexural strength (MPa)	250.8 ± 9	269 ± 25	0.152
Elastic modulus (GPa)	10.1 ± 2.5	10.3 ± 2.2	0.943
Flexural strain	0.032 ± 0.01	0.037 ± 0.01	0.354
Energy to ultimate load (toughness) (MPa)	4.7 ± 1.6 *	8.1 ± 1.8 *	0.003
**Property** **Demineralized Samples**	**Control** **(*n* = 9)**	**Treatment** **(*n* = 8)**	** *p* ** **-Value**
Max force (N)	19.48 ± 2.96	17.17 ± 3.6	0.167
Tensile strength (MPa)	3.4 ± 0.28 *	2.47 ± 0.44 *	0.001
Stiffness (N mm^−1^)	16.3 ± 4.1	13.8 ± 5.4	0.339
Strain at failure	0.40 ± 0.14	0.27 ± 0.1	0.057

* Indicates a significant difference (*p* < 0.05).

**Table 2 pharmaceuticals-18-00164-t002:** Summary of crystal size data of mineralized samples.

Crystal Size of Mineralized Samples	Control (*n* = 10)	Treatment (*n* = 10)	*p*-Value
Crystal size 002 plane (Å)	186.5 ± 30.6	195.1 ± 24	0.6
Crystal size 211 plane (Å)	50.9 ± 1.5 *	53.6 ± 3.4 *	0.037
Crystal size 310 plane (Å)	85.6 ± 16.5	102.3 ± 17.5	0.1

* Indicates a significant difference (*p* < 0.05).

## Data Availability

Data are contained within the article or Appendix A.

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
