# Peer review of "Prolonged Impact of Bisphosphonates and Glucocorticoids on Bone Mechanical Properties"

_pharmaceuticals, 2025, doi:10.3390/ph18020164_

Round 1

Reviewer 1 Report

Comments and Suggestions for Authors

The manuscript has significant shortcomings in explaining the results, and there is no correlation between the outcomes obtained from different characterization techniques. The conclusions are extremely brief, and the introduction lacks sufficient literature to clarify the purpose of this study. Additional important characterizations could be performed to improve the manuscript, as well as a comparison with existing literature. In this context, the article in its current state cannot be accepted; therefore, my decision is to reject it.

Comments on the Quality of English Language

The English can be improved

Author Response

Reviewer 1:

The manuscript has significant shortcomings in explaining the results, and there is no correlation between the outcomes obtained from different characterization techniques. The conclusions are extremely brief, and the introduction lacks sufficient literature to clarify the purpose of this study. Additional important characterizations could be performed to improve the manuscript, as well as a comparison with existing literature. In this context, the article in its current state cannot be accepted; therefore, my decision is to reject it.

Dear Reviewer 1,

Thank you for your comments. The Introduction section was adjusted to better reflect the context of the manuscript, and the Conclusions section was expanded. Additional changes were highlighted throughout the revised version. Furthermore, the manuscript was re-organized to 1.Introduction, 2.Results, 3.Discussion, 4.Materials and Methods, 5.Conclusions as requested by the Assistant Editor.

Reviewer 2 Report

Comments and Suggestions for Authors

The manuscript meets the journal’s remit.  The study aims to assess the impact of bisphosphonate and corticosteroid treatment on mature male rat long bone mechanical and biochemical properties using a range of outcomes.  The results show that there was an increase in cortical thickness but not cross sectional area in ZA/DX treated rats. Max force and toughness were also significantly increased in treated animals. Tensile strength decreased in treated groups.  This was associated with increased mineral size and collagen cross linking in the treated group.

The area is of interest and worthy of study. The introduction although relevant is rather vague at the moment. The study’s rationale would be improved by providing additional detail on specific instances where bisphosphonates and corticosteroids are prescribed together in males who have metastatic bone cancer.  

Line 84 “holyday” should be holiday.

The materials and method are described to an appropriate depth. The rationale for the exclusion of rats due to greenstick type fractures should be included in the method as half of the control cohort were excluded because of this.

The results are described well, and relevant statistical differences are noted between experimental and control groups.  Data is presented in figures and tables to a good standard.  The standard of English is of publication quality.

The major issue is the low sample size especially in the control group for the mineralised three point bending data. 50% of the animals were not included due to oblique fractures.  This is noted in the discussion, and although not reaching statistical significance is a major biological outcome of the work. How does this compare to the nature of atypical fractures noted in males who have been given ZA and Dex in combination?

The cellular basis of the observed changes should also be discussed.  What was leading to the increased cortical thickness in treated animals?  Is this reflective of an anabolic action or suppression of a catabolic change in controls?  Histomorphometric analysis of samples should be considered, what is the osteoclast and osteoblast areas at cortical envelopes?  Tartrate resistant acid phosphatase (osteoclast) and Goldner’s staining (osteoid) should also be considered. 

Author Response

Reviewer 2:

The manuscript meets the journal’s remit.  The study aims to assess the impact of bisphosphonate and corticosteroid treatment on mature male rat long bone mechanical and biochemical properties using a range of outcomes.  The results show that there was an increase in cortical thickness but not cross sectional area in ZA/DX treated rats. Max force and toughness were also significantly increased in treated animals. Tensile strength decreased in treated groups.  This was associated with increased mineral size and collagen cross linking in the treated group.

Dear reviewer 2,

Thank you for your comments to enhance the scientific merit of the study.

The area is of interest and worthy of study. The introduction although relevant is rather vague at the moment. The study’s rationale would be improved by providing additional detail on specific instances where bisphosphonates and corticosteroids are prescribed together in males who have metastatic bone cancer. 

The Introduction was adjusted to include additional details on the prescription of bisphosphonates and corticosteroids. The following text was added to the first paragraphs of Introduction

Bone metastases can be osteoblastic (e.g. prostate cancer in males), osteolytic (e.g. breast cancer in females and multiple myeloma in both males and females), or mixed. As a result, bisphosphonates have been widely used in cancer treatment to reduce the number of bone-related events and preserve of the bone integrity. Glucocorticoids are also used extensively in the treatment of cancer as an adjuvant therapy to reduce the side effects of cancer medications.

Line 84 “holyday” should be holiday.

Thank you!. The typo was corrected.

The materials and method are described to an appropriate depth. The rationale for the exclusion of rats due to greenstick type fractures should be included in the method as half of the control cohort were excluded because of this.

Thank you for your comment. The following text was added to the Materials and Methods section under “4.8. Statistical analysis” to explain the rationale for exclusion of samples with greenstick fractures.

Bone samples with oblique (greenstick) fractures were excluded from all properties listed in Table 1. This guaranteed an accurate comparison for the cortical thickness and ensured more conservative interpretation of results.

The results are described well, and relevant statistical differences are noted between experimental and control groups.  Data is presented in figures and tables to a good standard.  The standard of English is of publication quality.

Thank you for your comment.

The major issue is the low sample size especially in the control group for the mineralized three point bending data. 50% of the animals were not included due to oblique fractures.  This is noted in the discussion, and although not reaching statistical significance is a major biological outcome of the work. How does this compare to the nature of atypical fractures noted in males who have been given ZA and Dex in combination?

Thank you for your comment. Although this is an important outcome of the study, these samples were excluded to ensure more conservative interpretation of the results. As mentioned in a previous comment, the following text was added to the Materials and Methods section under “4.8. Statistical analysis” to explain the rationale for exclusion of samples with greenstick fractures.

Bone samples with oblique (greenstick) fractures were excluded from all properties listed in Table 1. This guaranteed an accurate comparison for the cortical thickness and ensured more conservative interpretation of results.

We also added the following text to the first paragraph of discussion to highlight the implications of atypical fracture noted in our experiment

More samples in our experimental group appeared to have transverse or short oblique fracture configuration, which is one of the major features of atypical fracture [21]. Although the nature of bone fracture is a secondary outcome in our study, it is consistent with the literature indicating that bisphosphonates and glucocorticoids are risk factors for atypical bone fracture [4]. Nevertheless, in the same study, it was reported that the risk of atypical fracture decreases rapidly after bisphosphonates discontinuation, which could explain that the type of fracture in our study did not reach statistically significant level. The incidence of atypical fracture could be also affected by gender [22]. Males had more atypical fracture than females in individuals younger than 49 year-old. However, females aged 80-90 year-old represented most cases of atypical fractures in individuals older than 50 year-old [22].

The cellular basis of the observed changes should also be discussed.  What was leading to the increased cortical thickness in treated animals?  Is this reflective of an anabolic action or suppression of a catabolic change in controls?  Histomorphometric analysis of samples should be considered, what is the osteoclast and osteoblast areas at cortical envelopes?  Tartrate resistant acid phosphatase (osteoclast) and Goldner’s staining (osteoid) should also be considered.

We agree with the reviewer that cellular and hisphmorphometric analyses are important considerations. However, the current study focused only on the mechanical properties of bone, and cellular and hisphmorphometric analyses are outside the scope of our study. Therefore, the following text was added to the last paragraph of the Discussion:

Furthermore, our study did not include cellular or histomorphometric analyses of bone. These analyses were previously described in the literature [30-32] and are important considerations for the interpretation of the results.

Reviewer 3 Report

Comments and Suggestions for Authors

This is a very interesting paper of basic science on rats treated with dexamethazone and zoledronic acid and measuring several bone properties. Even though corticosteroids traditionally have a significant negative effect on bones, it is important to know that the combination of steroids with bisphophonates from the beginning of treatment can have a protective effect.  

Regarding the greenstick fractures, as the authors correctly state at the end of the paper,  where they describe the limitations of the paper, the numbers are small to draw any significant conclusions. 

I would suggest comparing the treatment group with a group of rats treated only with dexamethazone, in order to see the real effect of cancer treatment, but also have another group that would use oral bisphosphonates.

I would also suggest extending a bit more the conclusions section, in order to include a suggestion of application of these findings to humans. 

Some minor typos:

line 84: correct holiday spelling

line 400: change 'remains' to 'seems to remain'

Author Response

Reviewer 3:

This is a very interesting paper of basic science on rats treated with dexamethazone and zoledronic acid and measuring several bone properties. Even though corticosteroids traditionally have a significant negative effect on bones, it is important to know that the combination of steroids with bisphophonates from the beginning of treatment can have a protective effect. 

Dear reviewer 3,

Thank you for your comments to enhance the scientific merit of the study.

Regarding the greenstick fractures, as the authors correctly state at the end of the paper,  where they describe the limitations of the paper, the numbers are small to draw any significant conclusions.

Thank you for your comment. Although this is an important outcome of the study, these samples were excluded to ensure more conservative interpretation of the results.  To clarify this point, the following text was edited and additional text was added to the last paragraph of the Discussion:

Even though our study was designed with sufficient sample size to address our main objectives, some secondary observation such as of oblique greenstick fractures led to samples being excluded from the study. This approach ensures accurate and conservative interpretation of the results. Larger sample sizes would be needed to fully explore the observation of the greenstick fracture. Furthermore, our study did not include cellular or histomorphometric analyses of bone. These analyses were previously described in the literature [30-32] and are important considerations for the interpretation of the results.

I would suggest comparing the treatment group with a group of rats treated only with dexamethazone, in order to see the real effect of cancer treatment, but also have another group that would use oral bisphosphonates.

We agree with the reviewer that additional groups are important consideration for this study to evaluate the impact of each drug individually on bone. Therefore the following text has been added to the last paragraph of Discussion

Additional studies with more experimental groups are needed to determine the impact of bisphosphonates and corticosteroids separately on bone. Future studies with a wider range of treatments, doses and endpoints would be needed to further improve our knowledge on the long term effects of the combine therapy of bisphosphonates with glucocorticoids.

I would also suggest extending a bit more the conclusions section, in order to include a suggestion of application of these findings to humans.

Thank you for your comment. The following text has been edited and added to the Conclusions.

Therefore, within the limitations of our study, the outcomes could indicate that administration of bisphosphonates to patients treated with high doses of corticosteroids could improve the mechanical properties of bone. This effect could persist even after discontinuation of the medications.

Some minor typos:

line 84: correct holiday spelling

line 400: change 'remains' to 'seems to remain'

Thank you for your comment. Both typos were corrected.

Round 2

Reviewer 2 Report

Comments and Suggestions for Authors

The reviewer thanks the authors for the changes they have made to their manuscript.